

# Silver-spoon effect in agricultural crop consumers: crop consumption enhances skeletal growth in sika deer

Ayaka Hata[1], Midori Saeki[1], Chinatsu Kozakai[1], Rumiko Nakashita[2], Keita Fukasawa[3], Yasuhiro Nakajima[4], Ryodai Murata[5], Yuki Harada[5] and Mayura B. Takada[5]

[1] Institute of Livestock and Grassland Science, National Agriculture and Food Research Organization, Tsukuba, Ibaraki, Japan
[2] Forestry and Forest Products Research Institute, Tsukuba, Ibaraki, Japan
[3] Biodiversity Division, National Institute for Environmental Studies, Tsukuba, Ibaraki, Japan
[4] Research Center for Advanced Analysis, National Agriculture and Food Research Organization, Tsukuba, Ibaraki, Japan
[5] Faculty of Science and Engineering, Chuo University, Bunkyo-ku, Tokyo, Japan

## ABSTRACT

Owing to agricultural expansion worldwide, agricultural crops can have major effects on the life history traits of wildlife. However, the functional role of crop consumption on the life history traits of long-lived mammals is seldom evaluated quantitatively. Body size is an important life history trait because it is directly related to fitness. In this study, we investigated the functional role of long-term crop consumption on skeletal growth of sika deer (*Cervus nippon*). Crop consumption accelerated skeletal growth of not only the consumer but also the next generation, and its effect differed by sex. In females, the degree of crop consumption produced maximum differences of about 1.4 years in the ages at which 98% asymptotic size was attained. Furthermore, there was a maximum difference of 1.5 times in the skeletal growth rate. On the other hand, crop consumption did not always affect skeletal growth in males. The degree of crop consumption by mothers generated a maximum difference of about 15% in the hind-foot length of their fetus. This study revealed that long-term crop consumption makes a difference in skeletal growth of deer at an individual level, even within the same population. Crop consumption by the mother has ''a silver-spoon effect'' on the next generation from the fetus stage.

# INTRODUCTION

Expansion of human activity in the Anthropocene has substantially altered 75% of the land surface of the world (*IPBES, 2019*). Agricultural expansion is the most widespread change in land use, with over one third of the terrestrial land surface being used for cropping or animal husbandry (*IPBES, 2019*). Numerous studies have reported the negative effects of agricultural land use on wildlife, such as biodiversity loss, habitat fragmentation, and modification of movement patterns (*e.g.*, *Kerr & Deguise, 2004*; *Dudley & Alexander, 2017*;

Corresponding author
Ayaka Hata,
hata.ayaka294@naro.go.jp

*Tucker et al., 2018*; *IPBES, 2019*). However, agricultural crops provide nutritional benefits to wildlife (*Oro et al., 2013*; *Hata et al., 2021*). Agricultural crops can have large effects on wildlife because the crops are nutrient dense and abundant (*Zweifel-Schielly et al., 2012*; *Lowry, Lill & Wong, 2013*; *Oro et al., 2013*; *Birnie-Gauvin et al., 2017*) and are constantly resupplied (*Lowry, Lill & Wong, 2013*; *Oro et al., 2013*). The acquisition of nutrient-dense food can not only improve nutritional condition but also alter the life history traits of animals (*e.g.*, *Cook et al., 2004*; *McLoughlin, Coulson & Clutton-Brock, 2008*; *Sorensen et al., 2009*). There are individual differences in crop consumption, even within a population (*Ditmer et al., 2016*; *Bonnot et al., 2018*; *Hata et al., 2017*; *Hata et al., 2021*), suggesting that the degree of change in the life history traits associated with crop consumption may vary among individuals in a population. Therefore, quantitative comparison of crop consumption and its effect on life history traits within a population is needed to better understand the population dynamics of animals living in landscapes that include agricultural lands, although such studies are scarce (*Hata et al., 2021*).

Body size is a fundamental factor in the life history of animals because it is directly related to measures of fitness, including reproductive success and survival (*Lindström, 1999*). Body size is determined by environmental factors through maternal and paternal effects during the early stage of development ("congenital factors" in this study), and the persistence and strength of the effects of the congenital factors are fluctuated by environmental factors during the subsequent growth stage ("acquired factors" in this study) (*Lindström, 1999*). Especially in long-lived animals, the magnitude of the effects of the acquired factors may vary with growth stage and sex. For example, in cervids, which are long-lived, sexually dimorphic mammals, females reach the asymptotic body size earlier than males (*Yokoyama, 2009*; *Hata et al., 2025*) and attain reproductive maturity when they reach a critical body size threshold (*Solberg et al., 2002*; *Miura & Tokida, 2009*; *Flajšman, Jerina & Pokorny, 2017*). Therefore, an earlier trade-off occurs for females than for males in energy allocation between growth and reproduction (*Douhard et al., 2013*). The duration of growth is longer in males than in females (*Yokoyama, 2009*; *Hata et al., 2025*). Males may adjust resource allocation in response to nutritional conditions to obtain a larger body because body size is important in the competition for breeding opportunities and has a stronger effect on reproductive success than for females (*Clutton-Brock, Harvey & Rudder, 1977*; *Clutton-Brock, Guinness & Albon, 1982*; *Newbolt et al., 2017*; *Nishimura & Tujino, 2020*). The strategy for allocating resource to body growth according to sex has been assessed under resource-limited conditions (*Festa-Bianchet et al., 2004*; *Simard et al., 2008*; *Douhard et al., 2017*), but it is not clear how long-term consumption of high-nutrient crops affects this strategy.

There is little knowledge about the long-term consumption of agricultural crops by mothers on growth of offspring (*i.e.*, maternal effect). The nutritional condition of the mother affects the growth of the fetus and newborn offspring (*Keech et al., 2000*; *Monteith et al., 2009*; *Scott et al., 2015*). Larger body at birth may increase individual fitness because the body size and mass of fawns affects survival rate, body size after maturity, and reproductive success (*Kruuk et al., 1999*; *Takatsuki & Matsuura, 2000*; *Cook et al., 2004*; *McLoughlin,*

*Coulson & Clutton-Brock, 2008*). Assessing the effect of long-term consumption of high-nutrient agricultural crops by mothers on the body size of their offspring can reveal whether the offspring have an advantage at the start of their life.

In this study, we investigated the effects of long-term crop consumption as both congenital and acquired factors on the skeletal growth of sika deer (*Cervus nippon centralis*). We defined the effects during the growth period from conception to birth as congenital factors and those during the postnatal growth period as acquired factors. Nitrogen stable isotope values of bone samples were used as an index of the relative dietary contribution of agricultural crops (*Hata et al., 2021*). We hypothesized that body growth would be affected more strongly by long-term consumption of nutrient-dense foods than by short-term consumption. We assessed the effect of the $\delta^{15}$N values of bone samples on each parameter of the skeletal growth curve, which was estimated based on the measurement data of hunted deer by sex. Moreover, we assessed the effect of the $\delta^{15}$N values of the mother's bone on the skeletal size of her fetus. Finally, we demonstrated the functional role of agricultural crops in the life history of animals living in landscapes that include agricultural land.

## MATERIALS & METHODS

Portions of this text were previously published as part of a preprint (*Hata et al., 2024*).

### Study area

We studied the sika deer population inhabiting eastern Nagano Prefectures in central Japan (Fig. 1). The landscape is an agriculture–forest mosaic, including broad-leaved trees such as *Juglans* sp., *Quercus crispula*, and *Cornus controversa*, and coniferous trees such as *Cryptomeria japonica* and *Chamaecyparis obtusa*. The mountainous areas are covered with broad-leaved trees such as *Q. crispula* and *Betula platyphylla*; coniferous trees such as *Larix kaempferi*, *Pinus densiflora*, and *Abies mariesii*; This area has also an alpine zone (*Institute for Biodiversity Research and Education Earthworm, 2014*). Agricultural land includes both crop fields and sown grasslands and accounts for 20.8% of the study area. In this area, deer frequently consume vegetables and pasture grasses (*Hata et al., 2019*; *Nagano Prefecture, 2016*; *Tsukada, Ishikawa & Shimizu, 2012*). The elevation of the study area ranges from 700 to 2,500 m. The average of maximum snow depth at the foothill of Mt. Asama was 21.7 cm in 2018–2023 (*Japan Meteorological Agency, 2024*). The deer density of the study area was estimated by the harvest-based model as 28.12–31.42 individual/km$^2$ in 2019 (*Nagano Prefecture, 2021*).

### Data collection

In the study area, 219 skulls (145 females and 74 males; Table S1) and 42 fetuses (16 females and 26 males; Table S2) of wild deer hunted by local hunters or in animal control culls were collected during 2018–2020 and 2023. Based on *Hata et al. (2021)*, we used the total length of the skull (TL; *Von den Driesch, 1976*) as an index of body size of deer, except for fetuses. For fetuses, we used the length of the hind foot as an index of body size (*Hudson & Browman, 1959*; *Takatsuki & Matsuura, 2000*). The skull and hind-foot measurements were performed with a digital caliper to the nearest 0.01 and 0.1 mm, respectively.

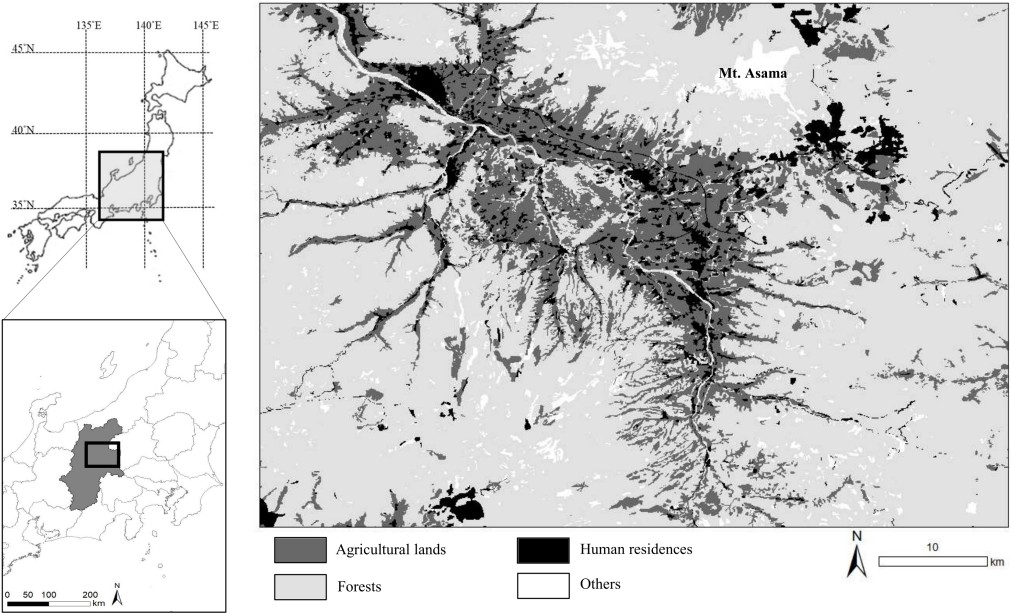

**Figure 1  Location of the study area in central Japan.** We collected specimens of wild sika deer hunted by local hunters or in animal control culls in this area during 2018–2020 and 2023. Base map was generated based on the Digital National Land Information (administrative district data) provided from Ministry of Land, Infrastructure, Transport and Tourism (https://nlftp.mlit.go.jp/ksj/gml/datalist/KsjTmplt-N03-2024. html). Vegetation map was generated from the 1/50,000 vegetation map provided from Biodiversity Center, Nature Conservation Bureau, Ministry of the Environment (http://gis.biodic.go.jp/webgis/sc-023.html).

The ages of the skull specimens that were collected from 2018 to 2019 were determined by using tooth replacement patterns (*Ohtaishi, 1980*). For specimens estimated to be ≥1 years old, the ages were determined from the cementum growth layers of the lower incisor root, a highly accurate method in ruminants known as the cementum age analysis (*Ohtaishi, 1980*; *Hamlin et al., 2000*). The all of skull specimens collected in 2020 and 2023 were estimated their ages by the cementum age analysis using the lower incisor root. The cementum age analysis was performed by Matson's Laboratory (Manhattan, MT, USA). Because the deer specimens were hunted in different seasons, we assumed June 1 as the birth date of all deer and then calculated the age in months based on the estimated age.

We estimated the gestation period for the fetuses. Generally, gestation age was estimated based on bone length or body weight (*Suzuki et al., 1996*; *Kobayashi et al., 2004*; *Yanagawa et al., 2009*). However, these methods can overestimate the gestation age of deer if the fetus grows faster and larger under high nutritional conditions. Although a previous study suggested that nutritional conditions have little effect on skeletal growth of fetus (*Kobayashi et al., 2004*), another study suggested that hinds with suboptimal nutritional condition during late pregnancy exhibited reduced fetal growth (*Scott et al., 2008*). In this study, we defined the period from the certain date of conception to the date the deer was hunted as the gestation period, and then we compared the hindfoot length of the fetuses. The representative conception date was assumed as October 15 because a reported

frequency distribution of conception dates for a deer population close to the studied population (Okunikko, central Japan) was highest in mid-October (*Iwamoto et al., 2009*). Thus, we defined the gestation period as from October 15 to the date the deer was hunted.

The $\delta^{15}N$ values in animal tissues are related to those in the animal's diet (*DeNiro & Epstein, 1981*). Therefore, differences in dietary contribution on food resources with different $\delta^{15}N$ values are thought to be reflected in the $\delta^{15}N$ values of consumers' tissues. To confirm the usefulness of $\delta^{15}N$ values as an index of the relative dietary contribution of agricultural crops, we collected food samples (wild plants and agricultural crops) which wild deer frequently consume in the study area (*Hata et al., 2021*). The previous study reported that deer inhabiting this area mainly consume graminoids (such as *Sasa* sp.) throughout the year in wild plants (*Takada et al., 2021*). Also, vegetables and pasture grasses are reported as frequently damaged agricultural crops by deer in this area (*Hata et al., 2019*; *Nagano Prefecture, 2016*; *Tsukada, Ishikawa & Shimizu, 2012*). We added some food samples on the food list which reported in previous study (*Hata et al., 2021*) (Table S3).

## Stable isotope analysis

To estimate the relative dietary contribution of crops in deer, we performed nitrogen stable isotope analysis using collagen and bulk bone fragments of the nasal turbinate taken from each skull specimen. Bone collagen has a slow turnover rate and thus provides dietary information spanning several years or the lifetime of the individual (*Hedges et al., 2007*; *Koch, 2007*; *Stenhouse & Baxter, 1979*). The $\delta^{15}N$ values of bone collagen reflect the long-term history of the relative dietary contribution of crops in deer (*Hata et al., 2021*). Because bone collagen extraction takes time and effort (>1 week to process each sample), we examined whether the bulk samples would provide similar information to the collagen samples in nitrogen stable isotope analysis. Bone comprises about 30% organic matrix and 70% minerals by weight (*Hall & Hall, 2021*). The organic matrix of bone is 90% to 95% collagen fibers, and the remainder is a ground substance (*Hall & Hall, 2021*). Because the $\delta^{13}C$ and $\delta^{15}N$ values of bone collagen primarily reflect the isotopic composition of dietary protein (*Ambrose & Norr, 1993*; *Tieszen & Fagre, 1993*), previous studies assessing diet and trophic level of animals using bone samples focused on isotopic values of bone collagen (*e.g.*, *Stevens, Lister & Hedges, 2006*; *Matsubayashi et al., 2014*; *Sakiyama et al., 2021*; *Hata et al., 2021*). The $\delta^{13}C$ values obtained from bulk samples and bone collagen samples cannot be treated equally because the $\delta^{13}C$ values in bulk samples are affected by carbon in hydroxyapatite carbonate, which is the main bone mineral (*Hall & Hall, 2021*), and carbon in the bone collagen, which is the main component of the organic matrix (*Hall & Hall, 2021*). Furthermore, the $\delta^{13}C$ and $\delta^{15}N$ values of bone samples collected from prehistoric animals are affected by diagenesis (*Yoneda, 2006*; *Kusaka et al., 2011*). However, the $\delta^{15}N$ values of bone collected from modern animals may not differ greatly between bulk and collagen samples because the main origin of nitrogen in bone is probably the organic matrix, which mainly consists of collagen.

To test the difference in the $\delta^{15}N$ values between the bone bulk and collagen samples of modern animals, we used wild deer samples ($n = 5$) collected in the same way as described above and captive deer samples (*C. nippon yesoensis*; $n = 4$). The deer were captured from

the natural environment and then kept at farm for about half a year in Hokkaido, Japan. In captivity, the deer mainly consumed pasture grasses. Bone samples were collected from the nasal turbinate taken from each skull specimen of all deer. Bone collagen and bulk samples were obtained through the following procedures.

The bone collagen samples were extracted following the methods described in *Hata et al. (2021)*. The specific methods are as follows: First, fat was removed by rinsing the bone samples with Folch solution (2:1 chloroform/methanol, v/v). The bone samples were soaked in one mol/L NaOH at −5 °C overnight. After NaOH was removed, the samples were lyophilized and ground into a fine powder. The powdered bone was decalcified by soaking in one mol/L HCl overnight and neutralizing with pure water. Following the acid pretreatment, the residues were gelatinized by heating at 90 °C for 12 h, after which the gelatin fraction was lyophilized. To obtain the bulk sample, fat was removed by rinsing the bone samples with Folch solution. The samples were lyophilized and then ground into a fine powder.

The bone collagen (0.3–0.4 mg) and bulk (1–1.5 mg) samples were weighed and then enclosed in a tin cup. Food samples were dried and ground into a fine powder, and then enclosed in a tin cup (grasses: 2.5–3.0 mg, woody barks and buds: 4.0–5.0 mg, agricultural crops: 2.0 mg). All samples were combusted in an elemental analyzer (FlashEA1112; Thermo Fisher Scientific, Waltham, MA, USA) connected to an isotope ratio mass spectrometer (DELTA V Advantage; Thermo Fisher Scientific), which was used to analyze the nitrogen isotope ratios. Nitrogen isotope ratios are expressed in delta ($\delta$) notation as parts per thousand (‰) relative to $R_{standard}$ as follows:

$$\delta(‰) = [(R_{sample}/R_{standard}) - 1] \times 1{,}000$$

where $R_{sample}$ and $R_{standard}$ are the $^{15}N/^{14}N$ ratios of the sample and the standard, respectively. The standard is the isotope ratio of atmospheric nitrogen (AIR). The analytical error for the isotope analysis was within 0.1‰ for $\delta^{15}N$.

## Statistical analysis

To assess whether the $\delta^{15}N$ values of agricultural crops are significantly higher than those of wild plants, we conducted Wilcoxon rank sum test.

Then, to assess whether the $\delta^{15}N$ values of bone bulk samples were equivalent to those of the bone collagen samples, we performed two one-sided tests to test for equivalence for each individual.

Several functions have been used to construct growth curves in animals (*France, Dijkstra & Dhanoa, 1996*). To select the best-fit functions for the TL of this deer population, we fitted the following three functions, which have been used to estimate growth curves of sika deer (*Hayashi et al., 2023*; *Suzuki et al., 2001*; *Uchida et al., 2001*).

The von Bertalanffy equation: $L(t) = a\left(1 - e^{-k(t-t(0))}\right)$

The logistic equation: $L(t) = \frac{a}{1 + be^{-kt}}$

The Gompertz equation: $L(t) = ae(-bk^t)$

Here, $t$ is the age in month of the deer, $L(t)$ is mean TL at the age in month $t$, $a$ is asymptotic TL, $b$ is the time scale parameter, $e$ is the base of natural logarithm, $k$ is growth

rate, and $t$ (0) is a fitting constant and is interpreted as the hypothetical age of an individual at zero TL, assuming the equation to be valid at all ages (*Eisen, Kang & Legates, 1969*; *Bartareau, Cluff & Larter, 2011*). All parameters were estimated using an R function for non-linear least square methods, nls. Because the growth curve of deer can vary according to sex (*Suzuki et al., 2001*), we generated models for females and males. To select the best-fit equation, we used Akaike's information criterion (AIC; *Burnham & Anderson, 2002*).

Based on the best-fit equation, we assessed the effect of crop consumption ($\delta^{15}N$ values of bone samples) on the TL growth curve. We constructed four models incorporating the linear predictor of $\delta^{15}N$ values for asymptotic TL ($a$) and growth rate ($k$) (model 1), growth rate ($k$) (model 2), asymptotic TL ($a$) (model 3), and none of the parameters (model 4). Because the growth curve of deer can vary according to sex (*Suzuki et al., 2001*; *Yokoyama, 2009*), we estimated models for females and males. All parameters were estimated using the nls() function. To select the best model, we used AIC.

Finally, we assessed whether a mother's crop consumption can affect the body growth of her fetus by linear model analysis. In the model, we set the hind-foot length as the dependent variable and the gestation period, sex of the fawn, $\delta^{15}N$ values of bone samples collected from the mother, and age in years of the mother as the independent variables.

The significance level was set as 5%. All statistical analyses were carried out using R for windows 4.4.0 (*R Core Team, 2024*).

## RESULTS

The $\delta^{15}N$ values of agricultural crops ($\delta^{15}N = 2.3 \pm 2.8‰$) were significantly higher than those of wild plants ($\delta^{15}N = -6.2 \pm 3.5‰$) ($p < 0.001$, Fig. 2 and Table S3).

The $\delta^{15}N$ values of collagen and bulk of bone samples were equivalent for all individuals when the allowable range was 1.5‰ (Fig. 3). Therefore, we treated the $\delta^{15}N$ values of bulk samples as the same as those of the collagen samples. The average $\delta^{15}N$ values of bone samples for female and male deer were 3.2‰ (range $-0.5‰$ to 7.5‰) and 2.8‰ (range 0.2‰ to 5.3‰), respectively (Fig. 4). All deer sample data are shown in Tables S1 and S2.

The logistic equation and the von Bertalanffy equation had the lowest AIC values for females and males, respectively. However, the $\Delta$AICs of all models were less than 2 for both sexes. Based on this result, we selected the logistic equation as the best fit for predicting the growth curve for both sexes (Table 1).

Based on the logistic equation, we assessed the effect of long-term crop consumption on the TL growth curve for each sex. Of the female models, model 2 had the lowest AIC, but the $\Delta$AICs of models 1 and 3 were less than 2 (Table 2). Based on the best model, we estimated growth curves incorporating the $\delta^{15}N$ values of the bone samples to assess the effect of the $\delta^{15}N$ values on skeletal growth (Fig. 5). We also showed other models in Fig. S1. For the female model, we estimated the age at which 98% of the asymptotic TL was attained with the highest (7.5‰) and lowest ($-0.5‰$) $\delta^{15}N$ values of bone samples as 2.9 and 4.3 years, respectively (Fig. 5). The predicted $k$ values (growth rate) were 0.09 with the highest $\delta^{15}N$ values and 0.06 with the lowest $\delta^{15}N$ values. Of the male models, model 1 had the lowest AIC, but the $\Delta$AICs of all models were less than 2 (Table 2). Because all male

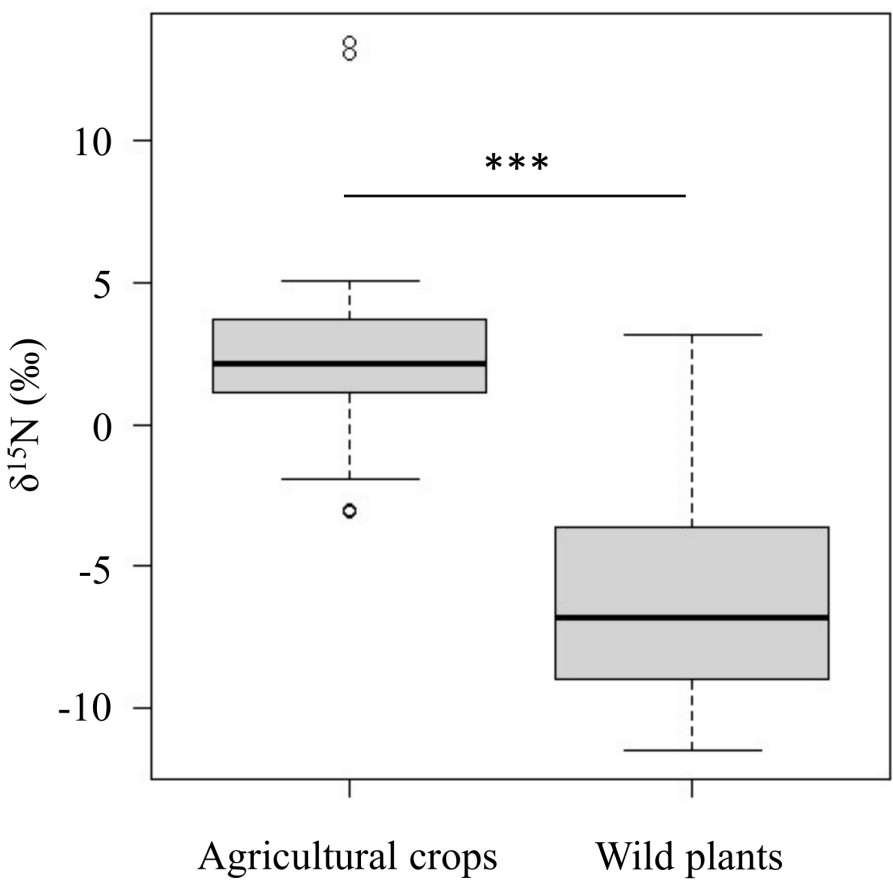

**Figure 2** Comparison of the $\delta^{15}$N values between agricultural crops and wild plants which wild deer frequently consume in the study area. Horizontal bars inside boxes indicate median values. Upper and lower ends of the boxes represent 25th and 75th percentile values, respectively. Asterisks (***) indicate a significant difference between foods ($p < 0.001$, Wilcoxon rank sum test).

models including null model had equivalent explanatory power, all models are shown in Fig. S2.

To assess whether a mother's crop consumption can influence the skeletal growth of her fetus, we generated a linear model. Positive relationships of the hind-foot length of the fetus were found with the gestation period, $\delta^{15}$N values of bone samples collected from the mother, and the age of mother (Table 3). The hind-foot lengths did not differ significantly with the sex of the fetus. We also estimated the hind-foot lengths at birth (231 days; *Suzuki, 1994*) with the 95th (4.9‰) and 5th (0.6‰) percentiles of the $\delta^{15}$N values of mother's bone samples as 22.0 and 20.1 cm, respectively. Moreover, we estimated the hind-foot lengths at birth with the highest (6.0‰) and lowest (−0.5‰) $\delta^{15}$N values of mother as 22.6 and 19.6 cm, respectively (Fig. 6).
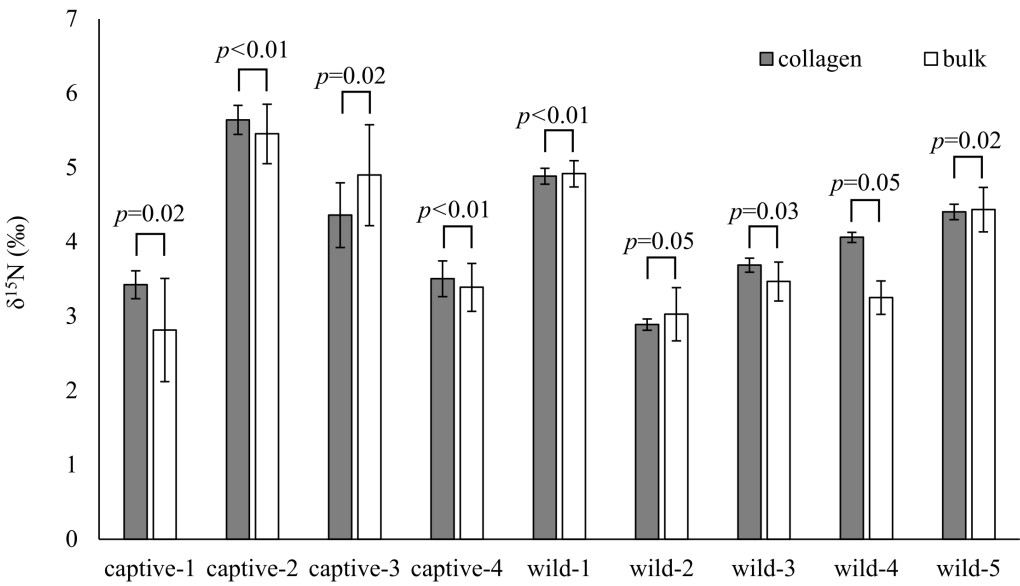

**Figure 3** **Comparison of the $\delta^{15}N$ values obtained from bone collagen (gray bars) and bulk (white bars) samples for each individual.** The horizontal axis shows the deer ID. "captive-**" and "wild-**" indicate captive and wild deer samples, respectively. Longitudinal bars indicate standard deviation. The $p$-values indicate the results of the equivalence test (two one-sided tests) with an allowable range of 1.5‰.

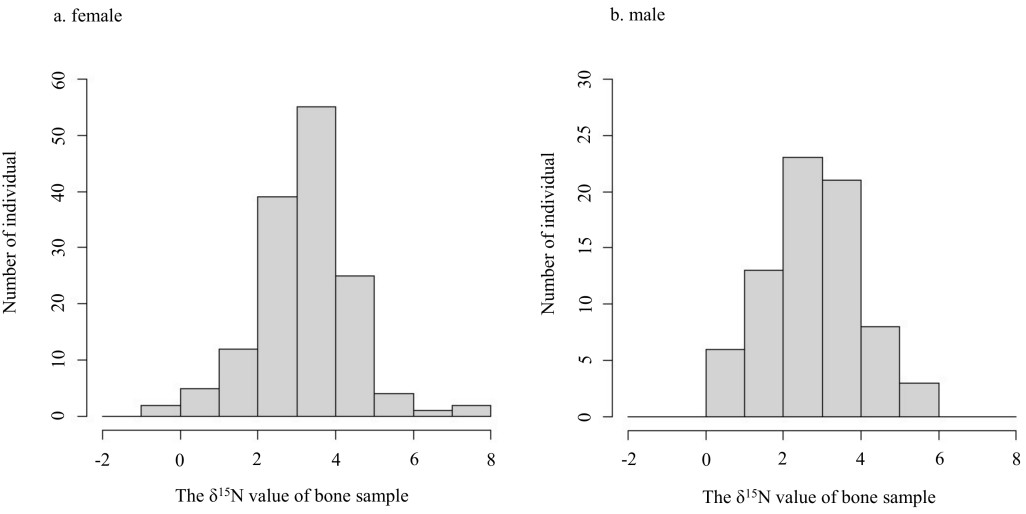

**Figure 4** **Histograms of the $\delta^{15}N$ values obtained from bone samples of female (A) and male (B) deer.**

## DISCUSSION

We found that agricultural crop consumption can act as both congenital and acquired factors that accelerate the body growth of deer, and its effect differed by sex. The variation in food habits produces differences in the speed of body growth within the same population.

**Table 1  Theoretical statistics and coefficients of three types of growth curves (Logistic equation, Gompertz equation, and von Bertalanffy equation) for the total length of the sika deer skull (TL).** The models were generated for females and males. Values in parentheses are standard errors.

| Model | AIC | ΔAIC | a | b | k |
|---|---|---|---|---|---|
| **Female (n=145)** | | | | | |
| Logistic equation | 1,024.21 | 0.00 | 269.53 (1.15) | 0.46 (0.03) | 0.07 (0.01) |
| Gompertz equation | 1,024.34 | 0.13 | 269.68 (1.17) | 0.40 (0.03) | 0.93 (0.01) |
| von Bertalanffy equation | 1,024.63 | 0.42 | 269.84 (1.19) | −16.64 (2.16) | 0.06 (0.01) |
| **Male (n=74)** | | | | | |
| von Bertalanffy equation | 537.25 | 0.00 | 287.77 (3.52) | −16.72 (3.42) | 0.06 (0.01) |
| Gompertz equation | 537.88 | 0.63 | 287.54 (3.49) | 0.43 (0.04) | 0.94 (0.01) |
| Logistic equation | 538.51 | 1.26 | 287.33 (3.47) | 0.50 (0.05) | 0.07 (0.01) |

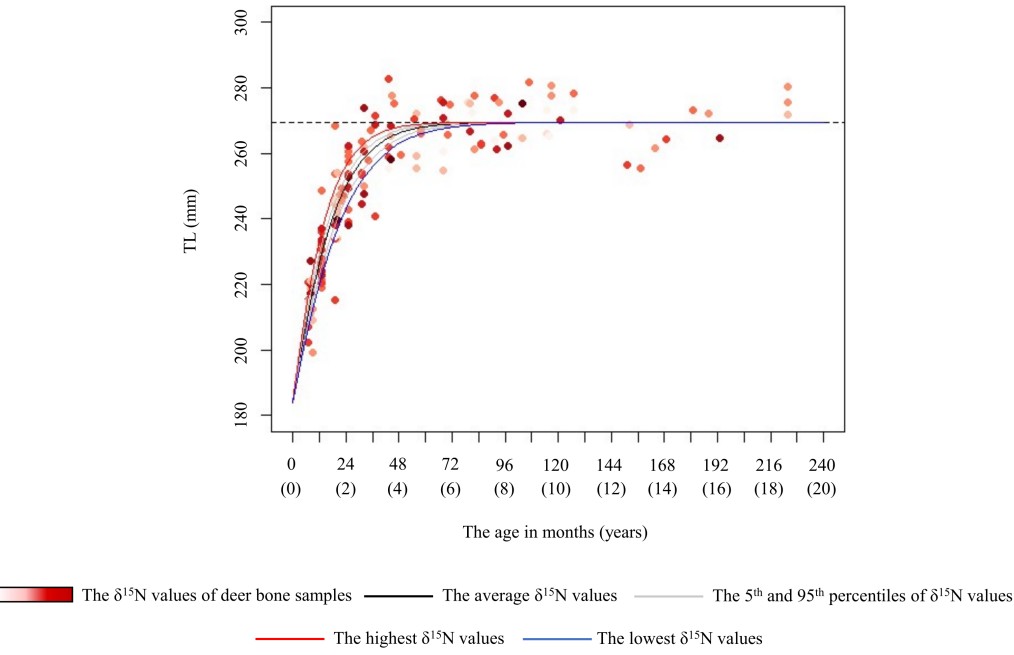

**Figure 5  Estimated growth curve for the total length of the female sika deer skull (TL) based on the best model (black line).** Grey curves were estimated by the model that incorporated the 5th (1.0‰) and 95th (4.9‰) percentiles of the $\delta^{15}$N values of bone samples as a parameter of the growth curve. Blue and red curves were estimated by the model that incorporated the lowest (−0.5‰) and the highest (7.5‰) $\delta^{15}$N values of the bone samples as a parameter of the growth curve, respectively. Numbers in parentheses on the horizonal axis indicate the age in years. The data points represent individual deer, with darker red indicating higher $\delta^{15}$N values. The horizontal dashed line indicates the asymptotic value estimated by the best model.

Although there is a gradient in the $\delta^{15}$N values of agricultural crops and wild plants, the $\delta^{15}$N values of agricultural crops were significantly higher than those of wild plants (Fig. 2). Because the $\delta^{15}$N values in animal tissues are related to those in the animal's diet (*DeNiro & Epstein, 1981*), differences in dietary contribution on these foods are thought to be reflected

**Table 2  Theoretical statistics and coefficients of the models assessing the effect of crop consumption ($\delta^{15}$N values of bone samples) on the growth curve of the total length of the sika deer skull (TL).** Based on the logistic equation, we constructed four models, which incorporated the linear predictor of $\delta^{15}$N values for growth rate (k) and asymptotic TL (a) (model 1), growth rate (k) (model 2), asymptotic TL (a) (model 3), and none of the parameters (model 4). b is the inflection point. $a_0$ and $a_1$ are the intercept and coefficient of the $\delta^{15}$N values of bone samples incorporating a, respectively. $k_0$ and $k_1$ are intercept and coefficient of the $\delta^{15}$N values of bone samples incorporating k, respectively. The models were estimated for females and males, separately. Values in parentheses are standard errors.

| Model | AIC | ΔAIC | $a_0$ | $a_1$ | $b$ | $k_0$ | $k_1$ |
|---|---|---|---|---|---|---|---|
| **Female (*n*=145)** | | | | | | | |
| model 2 $a_0/(1 + b * \exp(-(k_0 + k_1 * \delta^{15}N) * \text{month}))$ | 1,021.00 | 0.00 | 269.54 (1.13) | N/A | 0.47 (0.03) | 0.06 (0.01) | 0.00 (0.00) |
| model 3 $(a_0 + a_1 * \delta^{15}N)/(1 + b * \exp(-k_0 * \text{month}))$ | 1,022.87 | 1.87 | 266.15 (2.18) | 1.02 (0.56) | 0.47 (0.03) | 0.07 (0.01) | N/A |
| model 1 $(a_0 + a_1 * \delta^{15}N)/(1 + b * \exp(-(k_0 + k_1 * \delta^{15}N) * \text{month}))$ | 1,022.90 | 1.89 | 268.71 (2.92) | 0.25 (0.82) | 0.47 (0.03) | 0.06 (0.01) | 0.00 (0.00) |
| model 4 (Null) $a_0/(1 + b * \exp(-k_0 * \text{month}))$ | 1,024.21 | 3.21 | 269.53 (1.15) | N/A | 0.46 (0.03) | 0.07 (0.01) | N/A |
| **Male (*n*=74)** | | | | | | | |
| model 1 $(a_0 + a_1 * \delta^{15}N)/(1 + b * \exp(-(k_0 + k_1 * \delta^{15}N) * \text{month}))$ | 537.07 | 0.00 | 297.52 (6.49) | −3.96 (2.22) | 0.50 (0.04) | 0.04 (0.01) | 0.01 (0.00) |
| model 2 $a_0/(1 + b * \exp(-(k_0 + k_1 * \delta^{15}N) * \text{month}))$ | 537.21 | 0.14 | 288.21 (3.53) | N/A | 0.50 (0.04) | 0.06 (0.01) | 0.00 (0.00) |
| model 4 (Null) $a_0/(1 + b * \exp(-k_0 * \text{month}))$ | 538.51 | 1.44 | 287.33 (3.47) | N/A | 0.50 (0.05) | 0.07 (0.01) | N/A |
| model 3 $(a_0 + a_1 * \delta^{15}N)/(1 + b * \exp(-k_0 * \text{month}))$ | 538.90 | 1.83 | 285.16 (3.93) | 1.30 (1.05) | 0.50 (0.04) | 0.07 (0.01) | N/A |

in the $\delta^{15}$N values of deer's tissues. Also, it has been reported that the isotopic values in animal tissues change linearly in proportion to the food with different isotopic values (*Darr & Hewitt, 2008*). Actually, the $\delta^{15}$N values of bone samples varied greatly ($\delta^{15}$N= −0.5∼7.5‰; Table S1). This suggest that the higher $\delta^{15}$N values of consumers, the more likely they consumed agricultural crops. The result suggests that it is more appropriate to treat $\delta^{15}$N values as continuous values rather than setting a threshold value to identify whether the animal is a crop consumer.

The bone bulk samples provided $\delta^{15}$N values equivalent to those from the bone collagen samples for modern deer when the allowable range was 1.5‰. The variation of $\delta^{15}$N values was larger in bulk samples than in collagen samples (Fig. 3). A main reason proposed for

**Table 3  Results of the linear model analysis for the effect of each factor on the hind-foot length of the fetus.**

| Factor | Coefficient | t | P |
|---|---|---|---|
| Intercept | −11.75 (1.38) | −8.51 | <0.001 |
| **Gestation period** | 0.13 (0.01) | 20.62 | **<0.001** |
| Sex of fetus | −0.08 (0.50) | −0.17 | 0.87 |
| $\delta^{15}$N of mother | 0.46 (0.18) | 2.58 | **0.01** |
| **Age of mother** | 0.17 (0.07) | 2.32 | **0.03** |

Notes.
   Factors in bold are statistically significant.

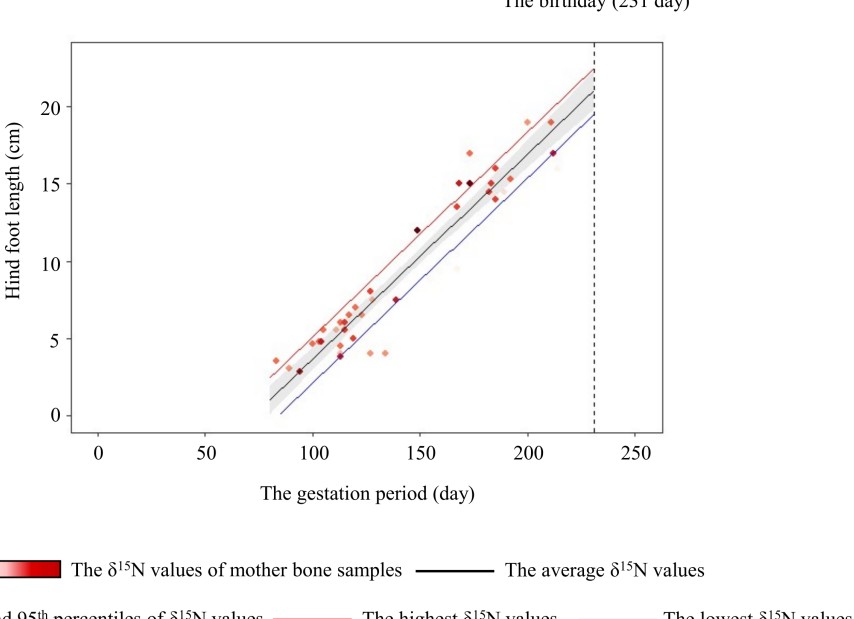

**Figure 6  Relationship between the hind-foot length and the gestation period of sika deer fawns.** The relationship estimated by the model is indicated by the black line. The 95% confidence interval is shown in grey. The relationships estimated by models that incorporated the highest (6.0‰) and lowest (−0.5‰) $\delta^{15}$N values of the mother are indicated by the red and blue lines, respectively. Each data point represents an individual fawn, with darker red indicating a higher $\delta^{15}$N value of the mother. Vertical dashed line indicates the average day of birth (231 days) of sika deer.

the large variation in bulk samples was that the bulk samples were less homogenized, than the collagen samples. Therefore, care should be taken to prepare the bone samples as a fine powder. Nevertheless, for analyzing modern deer, the $\delta^{15}$N values of the bone bulk samples provided sufficient information comparable to those of the bone collagen samples.

In female deer, crop consumption accelerated the skeletal growth rate for the growth duration (Fig. 5). Based on the best model for females, there was a maximum difference of about 1.4 years in the age at which they attained 98% asymptotic TL according to the degree of crop consumption (Fig. 5). Furthermore, we found that there was a maximum difference of 1.5 times in the skeletal growth rate. *Barr & Wolverton (2014)* reported

that the growth rate in body mass of white-tailed deer (*Odocoileus virginianus*) varies with food availability. Our study focused on skeletal growth rather than body mass, and consumption of nutrient-dense crops likely increased the skeletal growth rate in female deer. Growing larger in skeleton and body mass can be advantageous, resulting in higher survivorship, higher breeding success, and earlier reproduction (*Takatsuki & Matsuura, 2000*; *McLoughlin, Coulson & Clutton-Brock, 2008*; *Hata et al., 2021*). However, although the present models incorporating the $\delta^{15}N$ values and the asymptotic TL (model 1 and 3) had an explanatory power similar to the best model (Table 2), the effect of crop consumption on the asymptotic TL was unstable (Fig. 5 and Fig. S1). Although *Kaji, Koizumi & Ohtaishi (1988)* reported a reduction in the skeletal size with increasing deer density, *Simard et al. (2008)* reported that the skeletal size showed little change relative to body weight in response to food conditions. The effect of food availability on final skeletal size may occur under extremely severe food limitation. Because there were still sufficient natural food resources in this study area (*Hata et al., 2025*), the estimated asymptotic size may be the maximum size in this population, resulting from long-term local adaptions. The difference in the asymptotic size of the skeleton depending on crop consumption may be observed under more severe food limitation.

In male deer, crop consumption did not always affect skeletal growth. The null model (model 4) had explanatory power as well as the best model (model 1), which incorporated the $\delta^{15}N$ values for asymptotic TL and growth rate (Table 2, Fig. S2). Compensatory growth may be one reason why crop consumption did not always affect skeletal growth in males. Because the growth duration is longer in males than in females (*Yokoyama, 2009*; *Hata et al., 2025*), male body growth may be accelerated when food conditions improve after a period of food restriction (*Bohman, 1955*; *Douhard et al., 2013*). According to *Douhard et al. (2013)*, early environmental conditions directly affect adult body weight in female but not in male roe deer (*Capreolus capreolus*). This study focused on skeletal growth rather than body weight, but the long-term effects of the variation in nutrient acquisition on skeletal growth may have been mitigated for the long growth durations in male. Moreover, the sufficient food conditions in the study area (*Hata et al., 2025*) may have enabled compensatory skeletal growth in male for the growth duration.

The variation of crop consumption by mothers generated a maximum difference of about 15% in the hind-foot length of the fetus (Fig. 6). This is probably the first paper to quantify the effect of agricultural crop consumption on skeletal growth through generations in long-lived wild mammals. In large herbivores, offspring born to heavier individuals tend to have higher survivorship, pregnancy rate (*Keech et al., 2000*; *Douhard et al., 2014*), and lifetime reproductive success with precocious maturity (*McLoughlin, Coulson & Clutton-Brock, 2008*). Hindfoot length and body weight of fetuses are highly corelated with fetus age in mule deer (*O. hemionus*) and white-tailed deer (*Short, 1970*). *Takatsuki & Matsuura (2000)* showed that sika deer fawns with smaller hind-hoot length had a higher mortality rate in the first winter, with a difference in hind-foot length between the living and dead fawns of about 6%. Therefore, the estimated difference of about 15% in hind-foot length at birth in the present study was sufficient to raise the survival rates of

fawns. Crop consumption by the mother has "a silver-spoon effect" (*Grafen, 1988*) on the next generation from the fetus stage.

The present study revealed that the degree of long-term crop consumption makes a difference in skeletal growth of deer at an individual level, even within the same population. The accelerated growth rate of female deer was probably affected by a combination of the congenital and acquired factors. In contrast, crop consumption may only affect the skeletal growth of males as a congenital factor in the fetus stage. However, the difference in the effect size of each factor in promoting growth rate remains unknown. Because some studies have reported that the maternal condition affects the body mass and antler size of the offspring (*Keech et al., 2000*; *Monteith et al., 2009*), congenital factors are important in body growth. On the other hand, a long growth duration enables compensatory growth in deer affected by acquired factors in male deer (*Douhard et al., 2013*), as observed in this study. To determine the effect of acquired and congenital factors quantitatively, the effects of the changes in crop dependency during the lifetime of deer should be measured. A method has been developed to reconstruct the long-term chronological feeding history using the eye lenses of large mammals (*Miura et al., 2025*). This method would enable us to examine the size of the effect of crop consumption as acquired and congenital factors on skeletal growth. Furthermore, it is necessary to investigate whether the larger fetal size due to maternal crop consumption was caused by a higher growth rate or by an earlier conception date. In this study, we determined the conception date uniformly and examined the relationship between the gestation period and the hind-foot length. Focusing the formation status of the crown and root of the mandibular cheek teeth (*Kierdorf, Hommeksheim & Kierdorf, 2012*) may enable us to estimate the absolute fetal age. By estimating the absolute fetal age, we could clarify the mechanism by which the parental crop foraging affects fetal growth in greater detail.

It should be noted that the results are based on an indirect examination of the effects of crop consumption on deer skeletal growth. In this study, we used the $\delta^{15}N$ values of bone samples as an index of crop consumption of deer. The $\delta^{15}N$ values of agricultural crops were significantly higher than those of wild plants (Fig. 2 and Table S3), but there are variations in the $\delta^{15}N$ values even among agricultural crops. Therefore, deer with high $\delta^{15}N$ values might have consumed large amounts of crops, but it cannot be ruled out that deer consumed only small amounts of crops with particularly high $\delta^{15}N$ values. In other words, the $\delta^{15}N$ values does not necessarily reflect the actual amount of crop consumption. This study focused on bone sample, which is tissues that reflect relatively long-term dietary history, and therefore considered to reflect a certain degree of constant dietary history. Further experimental study will be useful to determine the direct effect of actual amount of crop consumption on skeletal growth. We also used deer samples which hunted during several years. There is little possibility of drastic changes in the availability of natural food resources and climate during the study period (*Japan Meteorological Agency, 2024*; *Hata et al., 2025*), but external factors that may affect body growth should also be considered in further study.

## CONCLUSIONS

Nevertheless, our results suggested that consumption of high-nutrient crops had a positive effect on skeletal growth of long-lived large mammals over generations. Further examination of the effects of fluctuations in survival and pregnancy rates associated with accelerated skeletal growth of individuals on population growth rates will help to elucidate the effect of crop consumption on population dynamics more accurately. To understand the effects of crop consumption on wild populations better, the negative aspects should also be elucidated; anthropogenic food consumption can have negative effects on populations through increased mortality (*Oro et al., 2013*; *Johnson, Lewis & Breck, 2020*) and infection risk (*Becker, Streicker & Altizer, 2015*) and reduced gut-microbiome diversity (*Gillman, McKenney & Lafferty, 2022*). The cost of the accelerated growth rate should also be considered. *Douhard et al. (2017)* reported that roe deer with a higher growth rate of body mass after weaning tend to lose weight more rapidly at older ages. To understand the overall effects of crop consumption on the life history traits of deer requires elucidation of the population dynamics of large mammals inhabiting landscapes that include croplands.

## ACKNOWLEDGEMENTS

We would like to thank the numerous hunters, Tsuyoshi Takeshita, Masaki Amari, Yuko Fukue, Takuhiko Suzuki, Shota Yamamoto and Hitomi Sato, who provided the sika deer specimens for this study. Most specimens were collected as part of the Komoro Wildlife Product Commercialization Project. We also thank Yukiko Matsuura (Hokkaido Research Center, Forestry and Forest Products Research Institute) for providing captive sika deer specimens; Yuto Suda (National Institute of Animal Health, NARO), Yukari Sakamoto (Institute of Livestock and Grassland Science, NARO), Yosuke Kogawa and Masaru Ogitsu (Technical Support Center of Central Region, NARO), Tatsuki Shimamoto and Rinko Iinuma and Syoko Tsukada (Nippon Veterinary and Life Science University), Sayuri Oyama, and Sachiko Manaka for their assistance with the sample treatment; Mugino O. Kubo (The University of Tokyo) for providing constructive suggestions; ThinkSCIENCE for English language editing.

### Funding

This research was supported by JSPS KAKENHI (Grant Numbers 19K20492, 22K05962, 23K17072, and 23K21779). The funders had no role in study design, data collection and analysis, decision to publish, or preparation of the manuscript.

### Grant Disclosures

The following grant information was disclosed by the authors:
JSPS KAKENHI: 19K20492, 22K05962, 23K17072, 23K21779.

### Competing Interests

The authors declare there are no competing interests.

## Author Contributions

- Ayaka Hata conceived and designed the experiments, performed the experiments, analyzed the data, prepared figures and/or tables, authored or reviewed drafts of the article, and approved the final draft.
- Midori Saeki performed the experiments, authored or reviewed drafts of the article, and approved the final draft.
- Chinatsu Kozakai conceived and designed the experiments, performed the experiments, authored or reviewed drafts of the article, and approved the final draft.
- Rumiko Nakashita conceived and designed the experiments, performed the experiments, analyzed the data, authored or reviewed drafts of the article, and approved the final draft.
- Keita Fukasawa conceived and designed the experiments, performed the experiments, authored or reviewed drafts of the article, and approved the final draft.
- Yasuhiro Nakajima performed the experiments, analyzed the data, authored or reviewed drafts of the article, and approved the final draft.
- Ryodai Murata performed the experiments, authored or reviewed drafts of the article, and approved the final draft.
- Yuki Harada performed the experiments, authored or reviewed drafts of the article, and approved the final draft.
- Mayura B. Takada performed the experiments, authored or reviewed drafts of the article, and approved the final draft.

## Animal Ethics

The following information was supplied relating to ethical approvals (*i.e.*, approving body and any reference numbers):

All of the deer samples used in this study were hunted by local hunters for the purposes of hunting or animal control culling under the "Act on the Protection and Management of Wildlife, and the Optimization of Hunting" established by the Japanese government. The authors obtained all of the deer samples from local hunters, through the appropriate procedures of affiliated institution. The authors were not directly touched in the hunting activities, and thus our experiments did not fall under the definition of animal testing at affiliated institution. Therefore, the authors do not have any animal care approval number and field permit number.

## Data Availability

The raw data and R code which used for statistical analyses are available in the Supplementary Files.

## Supplemental Information

Supplemental information for this article can be found online at http://dx.doi.org/10.7717/peerj.19836#supplemental-information.

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
