# Peer review of "Silver-spoon effect in agricultural crop consumers: crop consumption enhances skeletal growth in sika deer"

_PeerJ, doi:10.7717/peerj.19836_

## Round 0.1 · original submission · Major Revisions

Take careful note of the reviewers' comments, especially those pertaining to information that is lacking in the materials/methods and the conclusions drawn from the study.

Reviewer 1 ·

Basic reporting

This manuscript is well-written in professional English throughout. The references seem reasonable (however, I have some concerns related to the conclusions that are not independent of the references). The manuscript is appropriately structured, and the results are relevant; however, in my view, it is difficult to determine whether the authors' specific hypotheses related to 'crop consumption' were actually properly tested.

Experimental design

As I could tell, this research is novel and interesting, setting out to examine the relationship between crop consumption and growth in sika deer; growth of both the animals consuming said crops and of their offspring (developing foetuses). The authors conclude that crop consumption (as measured by 15N in bone tissues) enhances growth. The authors use a combination of experimental and mathematical approaches to understand the relationship between development time, crop consumption, and growth. While the findings are certainly interesting and likely have useful implications, my major concern is that the methodology employed does not test crop consumption in an entirely compelling way. Although I recognise that 15N in bone is a useful proxy for 15N in diet, based on the experimental design, it seems speculative that crop consumption necessarily explains 15N levels in bones. Firstly, which crops are the deer eating? None are specifically mentioned in the MS. How do we know that there is not a gradient in 15N values of the non-crop plants deer might eat that explains the levels of 15N values you detect in bones? Also, do we know how it scales? For example, a 50% crop diet vs a 30% crop diet vs a 10% crop diet — does this scale linearly in terms of the relationship between food and bone 15N?

It seems this is a difficult thing to control — do you have a 15N value from deer you are certain didn't eat any crops? As a control/reference?

Validity of the findings

While the findings are certainly interesting and likely have useful implications, I find the conclusions drawn, based on the results presented, to be not entirely convincing. I am convinced by the fact that 15N in food and 15N in bone are tightly related in a controlled way if animals are fed a consistent diet, but less in known under heterogenous diets (eating a bit of this, a bit of that). From what I can tell, this wasn't tested in any of the references either. If authors have this information, plus more information on the relationship between food 15N and bone 15N for deer for this study specifically, it would go a long way toward assuaging my concerns. Are there data on the levels of 15N in bones from deer that don't eat any crops?

I find this quite important because it is in the title of your manuscript. Crop consumption comes up twice in the title!

Additional comments

For figures 4 and 5, it is clear that the data points are deer, but what do the colours (on a scale from white to red) represent? 15N? Generally, the tables present a majority of the useful information, particularly related to 15N affecting growth (which is the whole point!). At the least, the figures need a legend, but to me, the k values for the blue and red curves in Figure 4 are the most interesting piece of data from the entire study. If you are suggesting an accelerated growth rate, it would be useful to see this numerically.

Reviewer 2 ·

Basic reporting

The authors of this paper should refrain from focusing on the impact of high-quality food (crop consumption) on the body weight of deer, given that they do not have actual body weight data for the analyzed (hunted) individuals. The same comment applies to the title. It may be worth considering replacing the phrase "body mass" with "skeletal parameters" or simply "skull size."

Experimental design

The analyses were conducted on individuals hunted in four different years (2018–2020 and 2023) and across various months. Did the authors take into account how these factors might have influenced the analysed parameters? It would be beneficial to include the year and month as random factors in the models.

I am uncertain about the extent to which the size of the deer skull is a reliable indicator of body weight, which can vary due to factors such as year, population density, and season. It is likely that variations in body weight caused by differing diets among deer may not necessarily be reflected in skull size.
I do not understand why the authors did not estimate the body weight of the fetuses. The hind foot length mentioned is not a reliable indicator of body weight in fetuses. The study by Takatsuki and Matsuura (2000), cited in the methodology, only shows a small coefficient of variation for hind foot length in fawns weighing 15-20 kg. Therefore, I recommend using a different coefficient to estimate body weight, preferably just the actual body weight of the fetus.

Validity of the findings

I suggest that the discussion focus primarily on the results obtained. It is important to note that the impact of food quality on the body weight of deer was not directly analysed, so any discussion of this impact should remain a hypothesis. The attached models indicate that high-quality food (crops) can influence the growth and development of deer during their early years. However, the authors should refrain from discussing the effect of good quality food on the body weight of fetuses, as they do not have relevant data to support such a claim.

Additionally, the discussion should address the research's limitations, which stem from the quality of the analysed material and the reliance on indirect indicators.

---

## Round 0.2 · Minor Revisions

It is recommended that the authors discuss (however briefly) that the conclusions are based on indirect results. This will further strengthen the paper and show that the authors are aware of the limitations of the study and future work that can be done to enhance it.

Reviewer 1 ·

Basic reporting

This manuscript is well-written in professional English throughout. Most of my concerns have been addressed. I am still unsure if the title is entirely appropriate given that a clear relationship between crop consumption and growth is not directly tested and you don't necessarily know that the highest 15N individual ate the most crops, maybe just ate a lesser proportion of the highest 15N-enriched crop. In my view: 'Silver-spoon effect in agricultural crop consumers: crop consumption enhances skeletal growth rates in sika deer', for example, would be a more appropriate title.

Experimental design

Authors have done their best to include additional data to support the relationship between 15N and crop consumption. Nevertheless, given limitations, the relationship between crop consumption and 15N in this system is difficult to elucidate given the range of 15N (ca. -3 per mille to 5 per mille) in the crop species alone. As the data are presented, I don't think 15N and crop consumption can be used interchangeably in this paper. What if you, based on the Darr and Hewitt paper, calculated the projected 15N diet values from bone values? This would give you a projected overall diet 15N level. Most of these values would be higher than any of the wild plant values, clearly showing crops were part of the diet. Any positive 15N diet values, for example, would guarantee crop consumption. Perhaps you could take these positive values and compare them to the most negative values (for which crop consumption would be lower) and directly compare growth rates between said groups?

Validity of the findings

Results are mostly valid, but I think the term crop consumption is overused. Some attempt to properly compare crop consumption vs. no crop consumption (as suggested above) is one way to get at this. I think this is critical as the word crop consumption appears twice in the title. Simply the fact that crops, generally, have higher 15N values than non-crops is insufficient.

Small comment on the figure legends. Authors refer to data points in (now) figure 5, as 'plots' which is confusing. The term 'plot', when used to describe data points, should be changed to 'data points' or something similar.

Additional comments

no comment

Reviewer 2 ·

Basic reporting

I am satisfied with most of the Authors' revisions. However, in the discussion and conclusions parts, the Authors should describe the limitations and weaknesses of these studies and emphasise that it should be borne in mind that the conclusions drawn in the paper are largely based on indirect results and therefore need to be verified experimentally in future studies.

Experimental design

It has not changed since the previous version and is largely based on indirect inferences. Importantly, the authors have changed their approach and now only describe the impact of agricultural food on skeletal development.

Validity of the findings

The authors should tone down some of their conclusions due to the lack of direct empirical evidence. Furthermore, it should be noted that there was considerable variability in the availability of natural food, environmental conditions, and climate between the years of the study.

---

## Round 0.3 · accepted · Accept

I have assessed the current version of the manuscript and believe that the authors have appropriately addressed all comments by reviewers.